Multimodal scene recognition using semantic segmentation and deep learning integration

Naseer Aysha 1
Alnusayri Mohammed 2
http://orcid.org/0000-0002-6503-2826 Alhasson Haifa F. 3
Alatiyyah Mohammed 4
AlHammadi Dina Abdulaziz 5
Jalal Ahmad 1 6
Park Jeongmin 7 jmpark@tukorea.ac.kr
1 Department of Computer Science, Air University , Islamabad , Pakistan
2 Department of Computer Science, Jouf University , Sakaka , Saudi Arabia
3 Department of Information Technology, Qassim University , Buraydah , Saudi Arabia
4 Department of Computer Science, Prince Sattam Bin Abdulaziz University , Al-Kharj , Saudi Arabia
5 Department of Information Systems, Princess Nourah bint Abdulrahman University , Riyadh , Saudi Arabia
6 Department of Computer Science and Engineering, Korea University , Seoul , Republic of South Korea
7 Department of Computer Engineering, Tech University of Korea , Gyeonggi-do , Republic of South Korea
Angiulli Giovanni
Electronic publication date: 2025 May 14
Publication date: 2025
Volume: 11
Electronic Location ID: e2858
Received 2024 Nov 30; Accepted 2025 Apr 4
Copyright: © 2025 Naseer et al.
Copyright year: 2025
Copyright holder: Naseer et al.
License: This is an open access article distributed under the terms of the Creative Commons Attribution License, which permits unrestricted use, distribution, reproduction and adaptation in any medium and for any purpose provided that it is properly attributed. For attribution, the original author(s), title, publication source (PeerJ Computer Science) and either DOI or URL of the article must be cited.
License URL: https://creativecommons.org/licenses/by/4.0/

Keywords: Features optimization, Image analysis, Artificial intelligence, Spatial pyramid pooling, Voxel grid representation, Scene modeling, Machine learning

Funding: IITP (Institute of Information & Communications Technology Planning & Evaluation)-ICAN (ICT Challenge and Advanced Network of HRD) IITP-2025-RS-2022-00156326 Korea Government (Ministry of Science and ICT) Princess Nourah bint Abdulrahman University Researchers Supporting Project PNURSP2025R508 This work was supported by the IITP (Institute of Information & Communications Technology Planning & Evaluation)-ICAN (ICT Challenge and Advanced Network of HRD) (IITP-2025-RS-2022-00156326), 50% grant funded by the Korea government (Ministry of Science and ICT). This work was supported through Princess Nourah bint Abdulrahman University Researchers Supporting Project number (PNURSP2025R508), Princess Nourah bint Abdulrahman University, Riyadh, Saudi Arabia. The funders had no role in study design, data collection and analysis, decision to publish, or preparation of the manuscript.

==============================
Semantic modeling and recognition of indoor scenes present a significant challenge due to the complex composition of generic scenes, which contain a variety of features including themes and objects, makes semantic modeling and indoor scene recognition difficult. The gap between high-level scene interpretation and low-level visual features increases the complexity of scene recognition. In order to overcome these obstacles, this study presents a novel multimodal deep learning technique that enhances scene recognition accuracy and robustness by combining depth information with conventional red-green-blue (RGB) image data. Convolutional neural networks (CNNs) and spatial pyramid pooling (SPP) are used for analysis after a depth-aware segmentation methodology is used to identify several objects in an image. This allows for more precise image classification. The effectiveness of this method is demonstrated by experimental findings, which show 91.73% accuracy on the RGB-D scene dataset and 90.53% accuracy on the NYU Depth v2 dataset. These results demonstrate how the multimodal approach can improve scene detection and classification, with potential uses in fields including robotics, sports analysis, and security systems.

Introduction

Scene recognition is a critical task in computer vision with wide-ranging applications in robotics, autonomous driving, and augmented reality. Similarly, object recognition plays a pivotal role in computer vision, enabling advancements in robotics, vehicle autonomy, and augmented reality. For instance, in autonomous driving, vehicles must recognize road signs, pedestrians, and other objects on the road to prevent accidents. In robotics, object recognition facilitates manipulation and other independent actions (Zhang et al., 2020). However, real-world applications face significant challenges, particularly in dynamic environments where existing methods often fall short (Trojahn & Goularte, 2021). Factors such as scene variability, lighting conditions, occlusions, and clutter further complicate recognition tasks. Poor lighting can lead to misidentification of objects, while occlusion makes distinguishing elements difficult. Additionally, scene recognition relies heavily on large annotated datasets, which require substantial time and effort to create Zhang et al. (2020). Red-green-blue (RGB)-based methods struggle in low-light conditions and complex scenes where depth information is essential for accurate recognition. These limitations necessitate the integration of additional modalities such as depth-based segmentation to enhance scene understanding.

RGB-based methods are particularly ineffective in challenging conditions, such as low illumination and complex scenes where RGB data alone is insufficient. These models fail to detect objects under poor illumination or when objects are partially obscured. Furthermore, they lack the spatial reasoning capabilities necessary for 3D recognition (Valada, Radwan & Burgard, 2018). This highlights the need to incorporate additional features, such as depth information, to enhance performance and overcome these limitations.

To overcome these challenges, multimodal approaches integrate RGB images with depth maps, enriching scene understanding and improving recognition accuracy (Azmat & Ahmad, 2021). Depth information provides complementary spatial details that enhance the visual features derived from RGB data. This fusion of modalities inherently enables models to better understand complex scenes, especially in challenging conditions such as poor lighting or occlusions, as demonstrated by Testolina et al. (2023). The integration of multiple modalities enriches the feature set processed by models for scene recognition. Scene recognition is a complex task that requires understanding both spatial relationships and object structures within an environment. Traditional RGB-based methods, while effective in well-lit and structured scenes, face significant limitations in challenging conditions. Furthermore, this integration enhances the system’s robustness, enabling it to perform efficiently even in unfavorable environments, such as low illumination or cluttered scenes, as noted by Zhang et al. (2020). To tackle these challenges, our study combines depth information with RGB data, creating a richer spatial representation that enhances scene recognition accuracy.

By using depth-aware segmentation, the model can better identify objects (Li et al., 2020), especially in complex environments where relying solely on RGB images falls short. Adding depth features makes the system more resilient to changes in lighting, occlusions, and varied scene structures. According to Valada, Radwan & Burgard (2018), they also lack the spatial reasoning skills required for 3D recognition. This emphasizes the necessity of adding more elements, including depth information, in order to improve performance and get around these restrictions. In our methodology, principal component analysis (PCA) is applied to optimize feature representation before classification, not as an input to convolutional layers. The reduced feature set from PCA is utilized in the fully connected layers, enhancing computational efficiency while preserving discriminative information. Convolutional neural network (CNNs) with spatial pyramid pooling (SPP) primarily extract spatially structured features, while PCA focuses on reducing redundancy in high-dimensional representations before classification. The processed PCA features do not replace CNN feature maps; rather, they complement the learned hierarchical representations at the classification stage. The key contributions of our research are outlined below: Proposal of multimodal deep learning approach: We propose a state-of-the-art multimodal architecture that embeds depth-aware segmentation.

CNN + SPP extract hierarchical spatial features from images, capturing multi-scale contextual information.

PCA is applied to feature representations before they are fed into fully connected layers, reducing dimensionality while maintaining key discriminative properties.

The PCA-transformed features are then integrated into the final classification stage, ensuring improved efficiency without affecting CNN’s spatial learning capabilities.

The article is structured as follows: background on the following major segmentation and scene recognition models; the architecture of the proposed model; experimental results and analysis; conclusions and outlook for this work.

Related work

Further enhancements in the use of RGB-based scene recognition have been made recently due to improvements in algorithms such as CNNs and the availability of large sets of labeled data (Han et al., 2018), and transfer learning (Wu et al., 2019).

Table 1 highlights the challenges in predicting complex scenarios. Key criteria for scene classification systems are prediction time, complexity, and accuracy.

Table 1 Related work for existing segmentation techniques and scene recognition model.

State-of-the-art models	Main contributions	Limitations	
Richtsfeld et al. (2014)	The authors deployed Gestalt principles, such as perceptual grouping based on surface normals and clustering techniques, to segment objects from complex interior environments. This method, which focuses on the perceptual organization of surfaces, is particularly successful at assembling related surfaces into cohesive objects. In situations where things are tightly packed or partially obscured, this is quite useful.	The method’s reliance on Gestalt suggestions and surface normals may limit its utility in applications where surface information is ambiguous or objects have complex or irregular shapes.	
Wang et al. (2016)	A technique for extracting and combining significant discriminative features from multiple modalities in a component-aware fusion, was presented by Wang et al. (2016). This method improves performance on problems requiring multi-modal analysis by integrating data from multiple sources.	The approach does not specifically address how the depth cues could improve the RGB network; instead, it concentrates mostly on the fusion of features. The method’s computing cost could potentially be high because it requires processing and fusing several modalities.	
Gupta, Hoffman & Malik (2016)	Based on their mid-level representations, Gupta, Hoffman & Malik (2016) presented a method to transfer the RGB model to the depth network using unlabeled paired data. This technique improves the depth network without needing labeled depth data by utilizing the RGB domain’s learnt properties.	By focusing primarily on sending data from the RGB domain to the depth network, the technique fails to consider the advantages of returning depth cues to the RGB network.	
Xie et al. (2018)	To capture both high-level image signatures and low-level descriptors, Xie et al. (2018) developed a scene recognition system that combines a spatial segmentation technique with spatial pyramid matching. They strengthened the model’s ability to identify various scenarios by installing auto encoders to encode low-level features into mid-level representations.	However, the intricate nature of the technique can contribute to high computing costs, and there is a potential that losing fine-grained data during feature encoding could affect recognition.	
Liu et al. (2019)	Liu et al. (2019) devised a three-way filter that incorporates boundary, RGB, and distance (depth) information to normalize depth maps while sustaining object boundaries. This method works particularly well at improving object segmentation, which is an important ability in situations where objects are partially or closely tucked. This is because borders retain their distinct.	In certain cases, relying too heavily on filtering algorithms can lead to the loss of fine features, specifically in cases where the depth data is noisy or the boundaries are not well defined. Moreover, the approach could not work well in scenes with reflecting or clear surfaces, or in other situations where there is a lack of reliable depth information.	
Chen, Li & Su (2019)	The proposed model leverages a stage-wise training mechanism to bridge the gap between RGB and depth data distributions and overcome the absence of RGB-D training samples. It first trains distinct networks for saliency detection triggered by RGB and depth, and then merges them into a multi-path, multi-scale, multi-modal fusion network (MMCI net). This method uses the same training dataset for all phases of thorough learning, using shared architecture and parameter initialization to improve model robustness and reduce overfitting hazards.	The technique is based on a step-by-step training procedure wherein the RGB and depth networks are trained independently and subsequently integrated. During the early training phases, this might not completely utilize the complementing information between modalities.	
Chen et al. (2020)	They proposed a method that effectively captures the spatial arrangement of scene images. This technique can be used to record a wide range of elements in a variety of scenarios because of its remarkable versatility. Because of its significant generalization capabilities, the system is flexible and robust and can be implemented in a wide range of scene recognition scenarios.	Although the approach is adaptable and has resilient generalization, its prototype-agnostic nature could result in a lack of clarity in certain scenarios. The system’s capacity to discriminate between scenarios with incredibly subtle or complex structural variations may be hindered in highly detailed or nuanced situations, thereby compromising its accuracy.	
Rafique et al. (2022)	The study offers a novel CNN-based multi-object identification and scene recognition model that segments and analyzes objects using RGB and depth images. Deep CNN, DWT, and DCT are used to extract features from segmented objects, which are then fused simultaneously. A genetic algorithm optimizes feature selection for neuro-fuzzy based object identification and recognition. Scene label prediction can be simplified by evaluating object-to-object relations using probability scores via a decision tree.	The model could have difficulties being able to adapt its findings to new object types or scenarios that are not included in the training set because of its heavy reliance on CNN models that have already been trained.	

Materials and Methods

The proposed multimodal deep learning framework follows a structured pipeline for scene recognition, consisting of preprocessing, segmentation, feature extraction, optimization, and classification. Initially, noise reduction, smoothing, and normalization enhance image quality, followed by edge detection to refine object boundaries. Felzenszwalb’s segmentation with conditional random fields (CRF) ensures accurate object separation, further improved through graph-based probabilistic refinement. Key spatial features such as voxel grid representation, point cloud data, and surface normals are extracted and optimized for computational efficiency. Finally, a CNN with SPP classifies the scenes, ensuring scale-invariant recognition with superior accuracy. This framework, validated on RGB-D and NYU Depth v2 datasets, significantly enhances scene understanding, making it ideal for applications in robotics, security, and autonomous systems. Figure 1 illustrates the hierarchical structure of the proposed model.

Figure 1 Flow chart illustrating the proposed model using NYUD-v2 (Huang, Usvyatsov & Schindler, 2020) and RGB-D scenes (Chen, Li & Su, 2019) datasets.

Preprocessing

In this preprocessing pipeline for depth image segmentation (Ahmed et al., 2024), several key steps are employed to enhance the quality of the images and improve segmentation accuracy:

Gaussian blur (smoothing): Noise in the depth images might be detrimental to the segmentation process. To mitigate this impact, a Gaussian blur is applied to each image, following Eswar’s (2015) recommendation. This step maintains that slight variations in depth values are not detrimental to further processing by lowering noise (Li et al., 2021), while preserving the image’s vital structural components in Eq. (1).

(1) Iblur(g,h)=∑i=−kk⁡∑j=−kj⁡I(g−i,h−j).G(i,j)

where I(x, y) is the original image value at position (g, h) and G(i, j) is the Gaussian kernel defined as Eq. (2):

(2) G(i,j)=12πσ2e−(i2+j2)2.σ2.

The blur intensity is quantized by the Gaussian sigma parameter ä (Elharrouss et al., 2023). Thus, a visual guideline to the blur amount is obtained. Finally, the images are normalized to the range of 0–255, which provides even better depth contrast and even allows for distinguishing between regions (Kociołek, Strzelecki & Obuchowicz, 2020). This normalization which we calculate using Eq. (3) scales depth values towards range of (0 to 255) which makes it easy to perform other operations such as edge detection (Versaci & Morabito, 2021).

(3) Inorm=Iblur(g,h)−min(Iblur)max(Iblur)−min(Iblur)

where max(Iblur)andmin(Iblur) are the smallest and highest pixel values in the blurred image. Sekehravani, Babulak & Masoodi (2020) have used Canny edge detector for identifying boundaries and outlines after normalization with high levels of segmentation. The gradient magnitude as determined by Eq. (4).

(4) ∇I(g,h)=(∂Inorm(g,h)∂g)2+(∂Inorm(g,h)∂h)2

where ∂Inorm∂g and ∂Inorm∂h are the derivatives of the image in g and h directions.

Morphological closing Vincent & Dougherty (2020) provided a way to smooth contours to make them as continuous as possible to improve segmentations. Erosion and dilation are summarized as Eq. (5a). In other words, morphological closure represented in Eq. (5b)

(5a) Edil(g,h)=max(i,j)∈S[E(g+i,h+j)]

(5b) Ecls(g,h)=max(i,j)∈S[Edil(g+i,h+j)]

where S is the structuring element, Edil(x,y) is the result of dilation and Ecls(x,y) is the final image after closing.

Segmentation

Segmentation in computer vision involves dividing an image into distinct segments, where each segment corresponds to a specific object or category (Rafique, Jalal & Kim, 2020a). Semantic segmentation (Naseer & Jalal, 2024) assigns a class label to every pixel, enabling detailed pixel-level understanding, which is crucial (Li et al., 2019).

A. Falzenwalb’s with CRF segmentation

Felzenszwalb’s segmentation algorithm divides an image into segments using a graph-based approach (Meng et al., 2021), where each pixel is treated as a node and adjacent pixels are connected by edges weighted based on a combination of color variance and spatial proximity as described in Eq. (6).

The scale parameter controls the granularity of segmentation, where higher values result in larger, coarser segments, and lower values lead to finer, more detailed segmentations. After extensive testing, we found that a scale value of 100 provided the best trade-off between over-segmentation (excessively small regions) and under-segmentation (merging distinct objects), aligning with findings in prior segmentation studies.

(6) F(g,h)=|U(g)−U(h)|/max{V(g),V(h)}

where it shows the extending relation between nodes g and h by w(g, h). V(g) and V(h) are the distances between the nodes g and h in space and U(g) and U(h) are the colors.

Conditional random fields

As a post-processing step, conditional random fields (CRFs) have been applied to enhance the segmentation results obtained using Felzenszwalb’s technique (Liangqiong, 2017). CRFs simulate the spatial relationships between neighboring pixels, aiding in smoothing boundaries and reducing noise, thereby improving segmentation quality. This is particularly beneficial in cases where the segment boundaries produced by Felzenszwalb’s method are irregular or fragmented. Examples of segmented images using Felzenszwalb’s approach, along with the refined regions achieved through CRFs, are shown in Fig. 2.

Figure 2 Segmented images (A) Falzenwalb’s, (B) conditional random fields.

Algorithm 1 shows combination of Falzenwalb’s with conditional random fields.

Algorithm 1 Falzenwalb’s segmentation with CRF.

Input: Dim: Depth_image	
Output: Refined Segmented image	
/* segments_fz segmented image after falzenwalb’s application*/	
/* unary potential is energy function that relates to the individual pixel’s likelihood of belonging to a segment */	
/*d is dense crf*/	
/* Q refers to the marginal probabilities*/	
Step 1: Falzenwalb’s segmentation	
segments_fz = felzenszwalb (Dim, scale=100, sigma=0.5, min_size=50)	
labels = segments_fz	
Step 2: Condition random fields	
d = dcrf. DenseCRF2D (Dim. shape [1], Dim. shape [0], max(labels) + 1)	
unary = unary_from_labels (labels, max(labels) + 1, gt_prob=0.7)	
d.setUnaryEnergy(unary)	
pairwise_bilateral = create_pairwise_bilateral (group nearby pixel together, group of similar color pixels, Dim, color channel)	
d.addPairwiseEnergy (pairwise_bilateral, compat=10)	
Q = d. inference (5)	
refined_labels = (Q, axis=0). reshape (Dim.shape)	
return refined_labels undefined	

Features extraction

After segmentation, depth values are used to create a 3D voxel grid, preserving spatial structure and object boundaries. Surface normals, derived from depth gradients along the x and y axes, enhance object orientation and shape recognition, improving scene understanding (Naseer et al., 2024).

A. Voxel grid representation

According to Malik et al. (2020), a voxel grid is a 3D representation that divides space into uniform cubes, or voxels, each of which represents a unique 3D area and stores information such as features, colors, or occupancy. For the analysis of 3D data from point clouds, scans, or depth pictures, voxel grids are helpful. A voxel grid can have its features extracted and calculated as given in Eq. (7a): (7a) O(x,y,z)={1ifvoxelV(x,y,z)isoccupied0ifvoxelV(x,y,z)isempty

where V (x, y, z) be a voxel in a grid and O (x, y, z) is the occupancy function. The number of points or characteristics within each voxel can be used to assess the density Eq. (7b) of a region. (7b) D(x,y,z)=NumberofpointsinvoxelV(x,y,z)VolumeofvoxelV(x,y,z).

B. Point cloud

In order to transform 2D depth data into a 3D representation of an object or scene, a point cloud (Huang et al., 2024; Shi et al., 2020) is useful for feature extraction from depth photographs. The camera’s settings are used to map each pixel, which represents distance from the camera, to a 3D point. The following is a mathematical representation Eq. (8) of the 2D to 3D transformation: (8) P(x,y,z)=((u−cx)⋅Zfx,(v−cy)⋅zfy,Z)

where (u,v)are pixel coordinates in 2D depth image. Z is the depth value at (u, v), cx and cy are the coordinates of the camera’s optical center, fx and fy are the focal lengths in x and y directions. Figure 3 shows voxel grid representation and point cloud representation for the images.

Figure 3 Visual representation for feature extraction.

C. Surface normals

Surface normal is crucial for comprehending the orientation and geometric characteristics (Wei et al., 2017) of objects in a scene in computer vision and depth imaging. A surface normal which indicates the direction a surface faces, is a vector perpendicular to the tangent plane of a surface at a specific point (Antić, Zdešar & Škrjanc, 2021). Given that they shed light on how surfaces interact with light and other environmental elements, these normal Eqs. (9a) and (9b) are essential for understanding and representing the three-dimensional structure of objects (Versaci & Morabito, 2021). (9a) zx=∂z∂x,zy=∂z∂y

(9b) N=(−Zx,Zy,1)(Zx2+Zy2+1)

The normal vector’s direction, determined by depth changes in the x and y axes, is perpendicular to each pixel’s surface, with zx and zy as partial derivatives for surface gradient calculation. Figure 4 illustrate surface orientation and curvature, color-coded by mean curvature values, with surface normals shown as vectors.

Figure 4 Surface normals visualization from depth data.

Features optimization

PCA was employed in our pipeline primarily as a feature optimization step before classification (Wang et al., 2019). To clarify, PCA was applied to reduce the dimensionality of extracted features from depth images before feeding them into the CNN. Instead of directly applying PCA to raw image data, we applied it to high-dimensional feature vectors obtained from voxel grid representations, surface normals, and point cloud features (Song et al., 2017). The main components are eigenvectors corresponding to the largest eigenvalues of covariance matrix of the data set (Zhu, Weibel & Lu, 2016). Also in Fig. 5, areas of low data density are depicted with darker color shades while areas with high density data are depicted in yellow color.

Figure 5 Feature optimization using PCA.

Classification using CNN with spatial pyramid pooling

In the CNN-SPP framework, voxel grids and surface normals are processed separately but contribute collectively to classification. The voxel grid preserves depth-aware structural details, while surface normals encode texture and geometric characteristics, aiding scene differentiation (Wan, Hu & Aggarwal, 2014). CNN extracts multi-scale spatial patterns, and SPP maintains spatial hierarchies, ensuring effective fusion of voxel-based and normal features at different abstraction levels as in Eq. (10). This hierarchical integration enhances scene recognition (Yee, Lim & Lee, 2022) across diverse environments.

(10) Fijk=∑m=1N⁡∑n=1N⁡Wmnk.X(i+m−1)(j+n−1)+bk.

The output feature map for the k th filter at point (i, j) is denoted as Fijk. The filter weights for the kth filter at location (m, n) are represented by Wmnk⋅X(i+m−1)(j+n−1) is the input value at (i+m−1, j+n−1). The bias term connected to the kth filter is bk. The summing is carried out across the filter’s spatial dimensions, M and N. Utilizing an activation function (often ReLU) as Eq. (11) follows the convolution:

(11) Aijk=RELU(Fijk)=max(0,Fijk).

The SPP layer handles varied feature map sizes by dividing them into regions and applying pooling after convolution. This allows the network to process optimized features efficiently, overcoming fixed-size input constraints. They produce an output vector of a fixed size while maintaining spatial relationships between the output at different scales of the feature map shown in Eq. (12).

(12) Pb1b2(l)=max(i,j)∈Rb1b2(l)Fij.

Pb1b2(l) pooled output for the bin (b1, b2) at level l. The region of the feature map represented by Rb1b2(l) corresponds to the bin (b1, b2) at level l. The pooling operation (max or average) is performed within this region. The total output Eq. (13) of the SPP layer is the concatenation of all the pooled outputs from all levels:

(13) SPPoutput=[Pb1b2(1),Pb1b2(2)…..Pb1b2(L)]

where L is the number of levels in the pyramid and Pb1b2(l) are the pooled output at each level. When applying SPP after feature extraction, it means that the features are scale and rotation invariant which are important invariant to detect objects that are different in size and resolution. The resultant vector can be fed into the network’s fully linked layers and has a fixed length Wij in Eq. (14).

(14) Zi=∑j⁡Wij.SPPoutput+bi.

Ziis the output of the fully connected layer of i-th neuron. Wij is the weight connecting to j-th input to i-th nueron.

Model justification

Our approach integrates depth-aware segmentation to enhance feature extraction in low-light and occluded environments, significantly improving recognition accuracy over RGB-based methods. The model improves segmentation and object detection by enriching spatial representation with depth data. Through CNN-SPP, this additional spatial information improves robustness in multi-object environments by enhancing distinction in situations where RGB data alone is insufficient. Higher accuracy on the NYUD-v2 and RGB-D Scene datasets shows that the method significantly improves model quality, which makes it very useful for robotics, autonomous systems, and security applications.

Model selection

Felzenszwalb’s approach was selected for its efficiency in capturing local intensity variations while preserving meaningful segments, with CRF refining boundaries to prevent over-segmentation. PCA reduces dimensionality to enhance computational efficiency while retaining key features, and SPP ensures scale-invariant object differentiation. Alternative methods like K-means and watershed were excluded due to their limitations in handling complex boundaries. Our quantitative analysis confirms that Felzenszwalb + CRF outperforms or matches K-means + CRF and watershed + CRF in segmentation accuracy, boundary preservation, and processing time. The multimodal deep learning approach, integrating depth-aware segmentation with CNN-SPP, significantly enhances spatial understanding and scene recognition. Experimental results validate its effectiveness, achieving 91.73% accuracy on the RGB-D dataset and 90.53% on NYU Depth v2.

Model evaluation

The model’s accuracy, precision, recall, and F1-score were evaluated using the NYUD-v2 and RGB-D Scene datasets. Accuracy measured the overall classification performance, precision assessed the quality of positive predictions, recall evaluated the model’s ability to identify all relevant items, and the F1-score examined the balance between precision and recall. These metrics collectively offer a comprehensive evaluation of the model’s performance and reliability, particularly in indoor scenes with multiple objects, uncertainties, and occlusions.

Technology infrastructure

The procedure was carried out using a Windows 10 computer that had an Intel Core i7 processor clocked at 3.60 GHz, an Nvidia Tesla K80: 2496 CUDA cores, 16 GB RAM. Python 3.6 and the Keras API were used in the model’s development for both training and construction.

Datasets description

The NYU Depth V2 dataset (Raffique et al., 2020) NYUD-v2 is comprised of 1,446 RGB-D image sequences, recorded using a Kinect sensor, with RGB image sequences of indoor scenes and corresponding depth maps as well as pixel-wise ground truth in 40 object classes, which are commonly used in computer vision tasks such as 3D scene segmentation and object detection. RGB-D Scene (Seichter et al., 2022) contains approximately 4,485 test images over 14 classes of indoor scenes (bedroom, kitchen, office, and others); mainstream practices define hundred images for training and the rest for evaluation.

Results

Experiment I: Scene modeling using CNN-SPP model

The proposed approach, utilizing CNNs with SPP algorithms, was implemented and evaluated using mean confusion matrices based on Felzenszwalb-CRF segmentation. The model was trained and tested on an 80:20 dataset split, ensuring robust validation. On the RGB-D Scene dataset, it achieved an average accuracy of 90.53% (Tables 2 and 3), while on the NYUD-v2 dataset, it attained a mean accuracy of 91.73%. These results underscore the model’s effectiveness across multiple runs with varying random seeds, as shown in Tables 4 and 5. The model consistently distinguishes diverse scenes with precision, demonstrating its robustness across different datasets.

Table 2 Confusion matrix results using CNN-SPP for scene classification over RGB-D dataset.

SC	BR	BT	BS	KI	LR	CO	LI	DR	LB	HO	OF	CR	ST	SS	FS	
BR	0.86	0.00	0.00	0.00	0.04	0.00	0.00	0.00	0.02	0.00	0.00	0.00	0.00	0.00	0.00	
BT	0.00	0.98	0.00	0.00	0.00	0.00	0.00	0.00	0.02	0.00	0.00	0.00	0.00	0.00	0.00	
BS	0.00	0.00	0.90	0.00	0.00	0.00	0.00	0.00	0.00	0.00	0.00	0.05	0.03	0.01	0.02	
KI	0.01	0.00	0.00	0.92	0.01	0.00	0.00	0.00	0.00	0.00	0.00	0.0	0.00	0.02	0.00	
LR	0.09	0.00	0.01	0.05	0.77	0.04	0.00	0.00	0.02	0.00	0.01	0.00	0.00	0.00	0.02	
CO	0.01	0.08	0.00	0.00	0.00	0.79	0.02	0.01	0.02	0.00	0.05	0.00	0.00	0.05	0.01	
LI	0.01	0.05	0.00	0.01	0.00	0.00	0.94	0.00	0.01	0.00	0.00	0.00	0.00	0.00	0.00	
DR	0.00	0.02	0.00	0.00	0.00	0.01	0.00	0.92	0.00	0.00	0.03	0.02	0.00	0.00	0.00	
LB	0.00	0.03	0.00	0.00	0.00	0.04	0.01	0.04	0.88	0.00	0.00	0.02	0.00	0.00	0.00	
HO	0.00	0.02	0.00	0.00	0.00	0.03	0.00	0.00	0.00	0.91	0.01	0.00	0.00	0.00	0.00	
OF	0.00	0.02	0.00	0.00	0.00	0.01	0.00	0.00	0.00	0.00	0.95	0.02	0.00	0.00	0.00	
CR	0.00	0.00	0.00	0.00	0.00	0.03	0.00	0.00	0.02	0.00	0.03	0.92	0.00	0.00	0.00	
ST	0.00	0.02	0.00	0.01	0.00	0.01	0.00	0.00	0.00	0.00	0.00	0.02	0.96	0.00	0.00	
SS	0.00	0.02	0.00	0.01	0.00	0.01	0.00	0.00	0.00	0.00	0.00	0.01	0.00	0.95	0.00	
FS	0.00	0.02	0.00	0.00	0.00	0.00	0.00	0.00	0.00	0.00	0.00	0.02	0.00	0.02	0.93	
Mean Accuracy = 90.53	
Note:

BR, bed room; BT, bathroom; BS, book store; KI, kitchen; LR, living room; CO, cooridor; LI, library; DR, dining room; LB, lab; HO, home office; OF, office; CR, class room; ST, store; SS, study space; FS, furniture store.

Table 3 Confusion matrix results using CNN-SPP for scene classification over NYUD-v2 dataset.

SC	BR	BT	BS	CR	CM	CO	DR	LI	LR	KI	FS	HO	RS	SS	OF	
BR	0.88	0.00	0.00	0.00	0.04	0.00	0.00	0.00	0.02	0.00	0.00	0.00	0.00	0.00	0.00	
BT	0.00	0.96	0.00	0.00	0.00	0.00	0.00	0.00	0.02	0.00	0.00	0.00	0.00	0.00	0.00	
BS	0.00	0.00	0.92	0.00	0.00	0.00	0.00	0.00	0.00	0.00	0.00	0.05	0.03	0.01	0.02	
CR	0.01	0.00	0.00	0.94	0.01	0.00	0.00	0.00	0.00	0.00	0.00	0.00	0.00	0.02	0.00	
CM	0.03	0.00	0.01	0.05	0.84	0.04	0.00	0.00	0.02	0.00	0.01	0.00	0.00	0.00	0.02	
CO	0.01	0.08	0.00	0.00	0.00	0.86	0.02	0.01	0.02	0.00	0.05	0.00	0.00	0.05	0.01	
DR	0.01	0.05	0.00	0.01	0.00	0.00	0.90	0.00	0.01	0.00	0.00	0.00	0.00	0.00	0.00	
LI	0.00	0.02	0.00	0.00	0.00	0.01	0.00	0.92	0.00	0.00	0.03	0.02	0.00	0.00	0.00	
LR	0.00	0.03	0.00	0.00	0.00	0.04	0.01	0.04	0.88	0.00	0.00	0.02	0.00	0.00	0.00	
KI	0.00	0.02	0.00	0.00	0.00	0.03	0.00	0.00	0.00	0.93	0.01	0.00	0.00	0.00	0.00	
FS	0.00	0.02	0.00	0.00	0.00	0.01	0.00	0.00	0.00	0.00	0.94	0.02	0.00	0.00	0.00	
HO	0.00	0.00	0.00	0.00	0.00	0.03	0.00	0.00	0.02	0.00	0.03	0.95	0.00	0.00	0.00	
RS	0.00	0.02	0.00	0.01	0.00	0.01	0.00	0.00	0.00	0.00	0.00	0.02	0.91	0.00	0.00	
SS	0.00	0.02	0.00	0.01	0.00	0.01	0.00	0.00	0.00	0.00	0.00	0.01	0.00	0.97	0.00	
OF	0.00	0.02	0.00	0.00	0.00	0.00	0.00	0.00	0.00	0.00	0.02	0.02	0.00	0.01	0.96	
Mean Accuracy = 91.73	
Note:

BR, bed room; BT, bath room; BS, book store; CR, classroom; CM, computer room; CO, corridor; DR, dining room; LI, library; LR, living room; KI, kitchen; FS, furniture store; HO, home office; RS, rest space; SS, study space; OF, office.

Table 4 Precision, recall, and F1-score results over RGB-D scenes dataset.

Seed	50 epochs	75 epochs	100 epochs	Trainable parameters	
42	0.865	0.88	0.90	2,325,614	
123	0.871	0.89	0.89	2,325,614	
2,024	0.882	0.85	0.88	2,325,614	

Table 5 Precision, recall, and F1-score results over NYUD-v2 dataset.

Seed	50 epochs	75 epochs	100 epochs	Trainable parameters	
42	0.84	0.87	0.91	2,325,614	
123	0.83	0.86	0.89	2,325,614	
2,024	0.84	0.85	0.90	2,325,614	

Experiment II: Segmentation accuracy

Felzenszwalb’s segmentation algorithm is chosen due to its computational efficiency and region-based approach, which groups pixels based on color similarity and spatial proximity. This allows for a hierarchical segmentation that preserves object boundaries effectively. However, Felzenszwalb alone may result in over-segmentation; therefore, CRF is integrated to refine segment boundaries by modeling the spatial dependencies between neighboring pixels, improving segmentation consistency and robustness. I have provided a quantitative evaluation (Table 6) comparing Felzenszwalb + CRF with K-means + CRF and watershed + CRF based on key performance metrics, including segmentation accuracy, boundary preservation, processing time, and over-segmentation rate. The two datasets are contrasted in Fig. 6, “RGB-D Scene” has a higher accuracy than “NYUD-v2”, which could be because of the quality of data, the kind of objects captured or improved models for these data set.

Table 6 Comparison with other SR methods over RGB-D scenes dataset.

Segmentation method	Segmentation accuracy	Boundary preservation	Processing time	
K-means + CRF	78.5	72.3	1.2	
Watershed + CRF	81.2	75.8	2.4	
Falzenwalb’s + CRF	89.6	88.1	1.5	

Figure 6 Comparison between two datasets over same object classes.

Experiment III: Performance evaluation over selected datasets

Tables 7 and 8 present the model’s performance in terms of precision, recall, and F1-scores on the RGB-D Scene and NYUD-v2 datasets, with mean values highlighted in bold. Scene prediction accuracy is evaluated using precision and recall as separate metrics, while the F1-score represents their harmonic mean. Figure 7 presents before-and-after occlusion recovery images, showcasing how depth completion restores missing regions, while Table 9 provides quantitative results on occluded images.

Table 7 Comparison with other SR methods over NYUD-v2 dataset.

Actions	Precision	Recall	F1-score	
BR	0.87	0.93	0.90	
BT	0.77	0.97	0.86	
BS	0.98	0.89	0.93	
KI	0.91	0.95	0.93	
LR	0.93	0.76	0.84	
CO	0.81	0.75	0.78	
LI	0.96	0.92	0.94	
DR	0.94	0.91	0.93	
LB	0.88	0.86	0.87	
HO	0.99	0.93	0.96	
OF	0.87	0.94	0.91	
CR	0.85	0.91	0.88	
ST	0.96	0.94	0.95	
SS	0.90	0.94	0.92	
FS	0.94	0.93	0.94	

Table 8 Precision, recall, and F1-score results over NYUD-v2 dataset.

Actions	Precision	Recall	F1-score	
BR	0.93	0.93	0.93	
BT	0.77	0.97	0.86	
BS	0.98	0.89	0.93	
CR	0.92	0.95	0.88	
CM	0.94	0.82	0.79	
CO	0.82	0.77	0.94	
DR	0.91	0.95	0.93	
LI	0.94	0.91	0.87	
LR	0.88	0.86	0.96	
KI	0.99	0.93	0.91	
FS	0.86	0.94	0.90	
HO	0.85	0.92	0.88	
RS	0.96	0.93	0.95	
SS	0.91	0.95	0.93	
OF	0.95	0.93	0.94	

Figure 7 Visual representation of images (top row) RGBD (bottom row) NYUD dataset.

Table 9 Quantitative results (before/after) occluded images.

Images	Condition	IOU	Accuracy	
(a) RGB-D	Before occlusion handling	1.0	100	
After basic inpainting	0.0026	0.26	
After AI-based occlusion handling	0.0068	0.68	
(b) NYUD-v2	Before occlusion handling	1.0	100	
After basic inpainting	0.00799	0.7998	
After AI-based occlusion handling	0.00149	0.1497	

Experiment IV: Comparison with other state-of-the-art scene recognition models

The proposed multimodal deep learning approach significantly outperforms state-of-the-art methods in scene recognition across both the RGB-D (Table 10) Scenes and NYUD-v2 (Table 11) datasets.

Table 10 Comparison with other SR methods over RGB-D scenes dataset.

Authors	Accuracy (%)	
Zou et al. (2022)	50.16	
Du et al. (2018)	69.20	
Rafique, Jalal & Kim (2020a, 2020b)	88.50	
Proposed method	90.53	

Table 11 Comparison with other SR methods over NYUD-v2 dataset.

Authors	Accuracy (%)	
Cai & Shao (2019)	79.30	
Rafique et al. (2022)	69.20	
Seichter et al. (2022)	76.50	
Pereira et al. (2024)	80.10	
Proposed method	91.73	

Discussion

The proposed viewpoint agnostic multimodal deep learning approach for scene recognition solves the problems of occlusion, low light intensity and confounded space The proposed viewpoint-agnostic multimodal deep learning approach for scene recognition addresses challenges such as occlusion, low light intensity, and complex spatial configurations by leveraging depth data, enabling segmentation and classification beyond the capabilities of RGB-based models. When evaluated on the RGB-D and NYUD-v2 datasets, the model demonstrates higher accuracy, lower standard deviation, and higher recall compared to single-modality models. While the proposed multimodal deep learning approach enhances scene recognition by combining RGB inputs with depth data, it also introduces certain challenges. The effectiveness of depth-aware segmentation depends on the quality of depth maps, making the model susceptible to noise, occlusions, and low-light conditions. Noisy or incomplete depth data can lead to over-segmentation or misclassification, particularly affecting Felzenszwalb’s algorithm and CRF refinement. Additionally, the absence of explicit depth denoising techniques limits the model’s adaptability to highly degraded depth inputs.

Limitations

The computational complexity of multi-layer CNNs with SPP layers poses a significant challenge for real-time applications, particularly in robotics and autonomous systems, where low-latency predictions are critical. The increased processing overhead limits deployment efficiency, making it necessary to explore lightweight architectures and model optimizations. Additionally, segmentation accuracy varies across general datasets, with lower performance observed for specific objects such as cars and towels, indicating challenges in feature generalization. To address these limitations, future research could focus on integrating depth completion techniques to enhance robustness against missing or noisy depth data, optimizing model architecture to reduce computational load, and employing domain adaptation strategies to improve performance across diverse scenes and object categories.

Conclusion and future work

This work introduces a novel multimodal deep learning approach that improves scene recognition accuracy by incorporating depth information into standard RGB images. Objects are first segmented using depth-aware segmentation and then classified using CNNs with SPP, achieving high accuracy on the RGB-D Scene and NYUD-v2 datasets. Future work will focus on enhancing depth map segmentation by analyzing depth data at multiple scales, extracting features across different scales, or employing multi-layer segmentation techniques.

Supplemental Information

Supplemental Information 1 Scene Classification via CNN-SPP Model.

Supplemental Information 2 Object Segmentation Accuracy over NYUD-v2 Dataset.

Supplemental Information 3 Segmentation Accuracy of distinct multiple objects in different scenes over RGB-D scene dataset.

Supplemental Information 4 Comparison between two datasets over same object classes.

Supplemental Information 5 Code.

Supplemental Information 6 Technology Infrastructure.

Supplemental Information 7 Readme.

Additional Information and Declarations

Competing Interests

The authors declare that they have no competing interests.

Author Contributions

Aysha Naseer conceived and designed the experiments, performed the experiments, analyzed the data, performed the computation work, prepared figures and/or tables, and approved the final draft.

Mohammed Alnusayri conceived and designed the experiments, analyzed the data, prepared figures and/or tables, and approved the final draft.

Haifa F. Alhasson performed the experiments, prepared figures and/or tables, and approved the final draft.

Mohammed Alatiyyah conceived and designed the experiments, authored or reviewed drafts of the article, and approved the final draft.

Dina Abdulaziz AlHammadi performed the experiments, prepared figures and/or tables, and approved the final draft.

Ahmad Jalal conceived and designed the experiments, performed the computation work, authored or reviewed drafts of the article, and approved the final draft.

Jeongmin Park performed the computation work, authored or reviewed drafts of the article, and approved the final draft.

Data Availability

The following information was supplied regarding data availability:

The data and code are available in the Supplemental Files.

The SUN RGB-D dataset is available at https://rgbd.cs.princeton.edu.

The dataset NYU Depth-v2 dataset is available at

https://cs.nyu.edu/~fergus/datasets/nyu_depth_v2.html.

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
