# Peer review of "Multimodal scene recognition using semantic segmentation and deep learning integration"

_PeerJ Computer Science, doi:10.7717/peerj-cs.2858_

## Round 0.1 · original submission · Major Revisions

The reviewers have provided valuable feedback. Please respond in an appropriate revision

Reviewer 1 ·

Basic reporting

The Introduction and Abstract effectively convey the paper's goal and address the existing research gap with clarity and precision. Author has done a great job.

To enhance the reader's understanding, it would be helpful to include a detailed explanation of the structure and layout of the two datasets used in the study. what is the target variable and what does the dataset look like overall?

For Table 2, consider expanding the information or providing an additional table to clarify the meaning of each label.

Figure 9 heading is incorrect and might need to be framed correctly. I believe it is the result of the NYUD dataset.

Experimental design

The methods described are sufficiently detailed to allow replication of the code and dataset. The process flow diagram or architectural diagram in Figure 1 effectively illustrates the data lineage, enhancing understanding.

Citations to relevant sources were appropriately provided, and the research was effectively benchmarked and compared with other relevant studies.

There was not much discussion around the reasoning behind model selection or method selection which could be added so that it enriches the paper and helps readers understand the rationale behind the model selection. Even if it is a hypothesis, readers would benefit from mentioning it.

Validity of the findings

The conclusion and future work sections were well-structured, effectively summarizing the research's essence and outlining how future work aims to address current limitations. The author did an excellent job crafting these sections.

The results were presented, with a logical flow in evaluating segmentation outcomes on the two datasets and discussing the accuracy, F1 score, and precision-recall metrics of the classification algorithm, which provided valuable insights.

However, the supplemental code provided for review did not align with the results discussed in the paper. Ensuring consistency between the code and the presented results would enhance the

Reviewer 2 ·

Basic reporting

Clarity and Professionalism of English Usage
The manuscript contains multiple grammatical errors and awkward phrasing that make it difficult to follow in several sections (e.g., lines 68–73, 94–100, 443–448). Key claims are often repeated without adding meaningful information, especially in the Introduction (lines 60–80). For example, the limitation of “RGB-based methods” is mentioned several times with no further elaboration.
Suggestion: Perform a thorough language review to correct grammatical errors, rephrase awkward sentences, and eliminate redundancy. Consider consulting a professional editing service.
Structure and Organization
The overall structure follows a standard format, but certain sections disrupt the flow. Furthermore, the Introduction does not adequately motivate the problem or present a clear hypothesis.
Suggestion: Reorganize subsections to improve readability. Ensure the Introduction presents a clear problem statement, research objectives, and hypotheses.

Experimental design

Flowchart and Methodology
The provided flowchart is ambiguous and raises several concerns about the methodology:
It is unclear how the segmentation outputs from Felzenszwalb’s method and CRF are transformed into voxel grids or surface normals.
The relationship between PCA and CNN + SPP is not explained. CNNs typically operate on spatially structured data, yet PCA reduces features to a low-dimensional vector, which is incompatible with CNN architectures.
Suggestion: The authors must provide detailed descriptions of:
How features are combined and used in the CNN + SPP pipeline.
Whether PCA features are reshaped for CNN compatibility or used in fully connected layers.
Hyperparameter Justification
Critical hyperparameters, such as the “scale” parameter in Felzenszwalb’s method or CRF settings, are not justified. The manuscript does not explain how these parameters were chosen or optimized.
Suggestion: Provide a rationale for hyperparameter choices or include an ablation study to demonstrate their effect on performance.
Dataset Splits
Furthermore, dataset splits (training/validation/testing) are not described.
Suggestion: Clearly describe the train–test splits and whether cross-validation was used to ensure reproducibility.
Occlusion Handling
The manuscript emphasizes occlusion as a key challenge but does not provide any experiments or results explicitly testing performance under occlusion. The flowchart and methodology do not explain how depth data specifically aids in occlusion handling.
Suggestion: Conduct dedicated experiments to evaluate the model’s performance under different occlusion levels. Provide visual examples (before/after) and quantitative results (e.g., IoU, accuracy) on occluded subsets of the dataset.

Validity of the findings

Authenticity and Novelty of the Results
The claimed accuracies (94.20% on RGB-D and 92.10% on NYUD-v2) are unusually high compared to state-of-the-art methods, yet no evidence of cross-validation or rigorous comparisons is provided. The methodology lacks transparency, making it difficult to assess the validity of these results. The mismatch between the reported mean accuracy (81.90%) in Table 3 and the claimed accuracy (92.10%) suggests potential inconsistency or selective reporting of results.
Suggestion: Include results from multiple runs, report random seed variability, and compare against strong baselines using the same splits.
Integration of Features
The manuscript claims to use Felzenszwalb’s method and CRF to produce voxel grids and surface normals as features. However, the process for generating these features is unclear:
a. How are voxel grids and surface normals derived from segmentation outputs?
b. How are these features fused or combined for downstream processing in CNN + SPP?
Suggestion: Provide a detailed explanation of feature extraction and fusion. If voxel grids and surface normals are processed separately, explain how they contribute to the final classification.
PCA in the Pipeline
PCA is used for "Feature Optimization" but is not traditionally compatible with CNNs due to its vectorized output. The manuscript does not clarify how PCA features are fed into the CNN.
Suggestion: Please give more details on them.
Discussion of Limitations
The manuscript briefly mentions limitations (lines 439–453) but fails to critically evaluate the challenges, such as:
a. How the model performs with noisy or incomplete depth data.
b. Generalizability to outdoor scenes or cluttered environments.
Suggestion: Expand the discussion of limitations and provide concrete steps for addressing these challenges in future work.

Additional comments

1. Repetition and Redundancy:
The manuscript repeatedly mentions the weaknesses of RGB-based methods without adding meaningful insights. This redundancy detracts from clarity.
Suggestion: Focus on explaining how the proposed multimodal approach overcomes these weaknesses instead of reiterating them.
2. Flowchart:
The flowchart lacks clarity on how features are extracted and combined.
Suggestion: clarify data flow and feature integration.
3. Fusion of Segmentation Outputs:

The proposed combination of Felzenszwalb’s method and CRF for segmentation is unconventional. However, the manuscript does not provide sufficient justification for why this combination is chosen over other segmentation techniques.
Suggestion: Justify the choice of Felzenszwalb + CRF and compare it with alternative segmentation methods.

---

## Round 0.2 · accepted · Accept

Dear Authors,

Your paper has been revised. It has been accepted for publication in PEERJ Computer Science. Thank you for your fine contribution.

Reviewer 1 ·

Basic reporting

The review feedback implementation looks good!

Experimental design

The review feedback implementation is supported with reasonable explanation!

Validity of the findings

Experiment evaluation and Conclusion feedback are accomodated.